# Additively-manufactured monocrystalline YBCO superconductor

Dingchang Zhang [1] ✉, Cristian Boffo [2] & David C. Dunand [1] ✉

Single-crystal microstructures enable high-performance $YBa_2Cu_3O_{7-x}$ superconductors which are however limited to simple shapes due to their brittleness. Additive manufacturing can fabricate $YBa_2Cu_3O_{7-x}$ superconductor with complex shapes, albeit with a polycrystalline microstructure. Here, we demonstrate a route to grow single-crystals from 3D-ink-printed, polycrystalline, sintered superconducting $YBa_2Cu_3O_{7-x}$ (YBCO or Y123) + $Y_2BaCuO_5$ (Y211), manufacturing objects with complex architectures displaying both high critical current density ($J_c$=2.1 × $10^4$ A·cm$^{-2}$, 77 K) and high critical temperature ($T_c$= 88-89.5 K). An ink containing precursor powders ($Y_2O_3$, $BaCO_3$, and CuO) is 3D-extruded into complex geometries and then reaction-sintered to obtain polycrystalline Y123 + Y211. A seed is then utilized to transform these 3D-printed parts from polycrystal to monocrystal via the melt growth method. The geometric details of 3D-printed parts survive the process without slumping, sagging or collapse, despite the long-term presence of liquid above the peritectic temperature. Origami structures can be created by sheet folding after 3D-printing. This additive approach enables the facile fabrication of superconducting devices with complex shapes and architectures, such as advanced undulator magnets to generate synchrotron radiation and microwave cavities for dark-matter axion search. This work highlights the potential of additive manufacturing for producing monocrystalline cuprate superconductors and opens the door to additive manufacturing of other monocrystalline functional ceramic or semiconductor materials.

$YBa_2Cu_3O_{7-x}$ (YBCO or Y123) is the original high-temperature cuprate oxide with a superconducting critical temperature ($T_c$~93 K) well above the boiling point of liquid nitrogen (77 K)[1], enabling various transformational applications, both for existing (e.g., magnetic resonance imaging, nuclear magnetic resonance) and future (e.g., energy storage, fusion power plants[2], maglev trains)[3]. Additive manufacturing of monocrystalline YBCO superconductors not only offers greater design flexibility for existing applications but also paves the way for new innovations. The architectural freedom of additive manufacturing enables optimized magnetic field distribution, efficient cooling to sustain the superconducting state, and a lightweight structure for levitation applications. Additive manufacturing of ceramics, like YBCO,

currently produces only a polycrystalline microstructure when followed by conventional sintering. However, grain boundaries are weak links for current transport in YBCO, principally because the coherence length (related to the characteristic size of a Cooper pair) of YBCO is too short to allow supercurrent to span over the structural disorder caused by a grain boundary[4]. Therefore, YBCO with a polycrystalline microstructure typically has a low critical current density (~5 × $10^1$ A·cm$^{-2}$, 77 K, zero field)[5] due to the presence of numerous grain boundaries. In contrast, forming a single-crystal microstructure effectively eliminates these grain boundaries, leading to a significant increase in critical current density (~4.5 × $10^4$ A·cm$^{-2}$, 77 K, H||c, zero field)[6]. Therefore, YBCO superconductors are mainly used as coated

[1]Department of Materials Science and Engineering, Northwestern University, Evanston, IL 60208, USA. [2]Fermi National Accelerator Laboratory, Batavia, IL 60510, USA. ✉e-mail: dingchangzhang2020@u.northwestern.edu; dunand@northwestern.edu

tapes/wires where the thin YBCO film is epitaxially grown on a biaxially textured conductor substrate[7] which can achieve a high critical current density (more than $3 \times 10^6$ A/cm$^2$ at 77 K [3]). However, these ceramic-like tapes cannot be bent over a radius below 2–10 mm without degrading their critical current[8], nor can they be shaped in complex 3D-objects. Furthermore, even for superconducting tapes, joints between strands to achieve longer length or larger width induce residual resistance[9]. This severely limits applications where complex 3D architectures are needed, such as microwave cavities with minimal gaps between tapes[10] and undulator with short periods[8]. In bulk form, YBCO (specifically: Y123 + $Y_2BaCuO_5$ (Y211)) single crystals (single grain or single domain) can be grown by the top-seeded melt growth or top-seeded infiltration growth method[3]. These bulk YBCO single crystals can trap large magnetic fields[11] and can potentially substitute iron-based permanent magnets, e.g., for superconducting motors[12] or generators[13]. However, they are currently limited to simple cylinder or cuboid shapes.

3D-ink-printing, an additive manufacturing (AM) method, extrudes a powder-loaded ink, layer by layer, and the resulting green part is then sintered to achieve densification[14,15]. The isothermal sintering and subsequent slow cooling prevent the formation of thermal-shock induced cracks in ceramics, which are frequently observed in laser-beam-based AM methods[16]. Also, ink can utilize submicron and nano-particles of various shapes and size distribution, unlike powder-bed-based AM methods (binder/ink jet printing[17]) for which powder flowability is limiting[18]. Previously, only polycrystalline Y123 superconductors have been fabricated by 3D ink-printing[5,19–24]. These polycrystalline microstructures are unable to create/trap a strong magnetic field or achieve heavy-load levitation.

Here, we develop and demonstrate a approach, where a single crystal is grown from a 3D-ink-printed Y123 + Y211 architectures with an excellent shape fidelity, achieving a nearly hundred-fold improvement in critical current density as compared to prior polycrystalline Y123 objects created by AM.

## Results and discussion
### Sintering
A powder-loaded ink - containing a blend of submicron $Y_2O_3$ + $BaCO_3$ + CuO precursor powders (Fig. S1) with a binder (PLGA), solvent (DCM), plasticizer (DBP), and surfactant (EGBE) - is extruded layer by layer from a 250 µm diameter nozzle to fabricate complex parts such as a toroidal coil, as shown in Fig. 1a. The rapid solvent evaporation after ink extrusion causes the binder to precipitate immediately after extrusion thereby increasing the strength of the deposited material, unlike non-evaporating ink systems. As a result, this ink system does not show slumping and sagging of the printed filaments[25] and thus enables them to partially overhang and form the arcs in the toroidal coil without auxiliary supports. After printing, a relatively dense microstructure is needed to keep the integrity of the printed structure during the subsequent single-crystal growth step. Also, high densities minimize porosity near the growth front of the single crystal, enabling fast single-crystal growth. However, the powder-loaded ink consists of loosely packed powders, making densification challenging. The composition targeted - pure $YBa_2Cu_3O_{7-x}$ (Y123) - is first used to study the synthesis and sintering behavior. As shown in Fig. 1b, the relative mass of the extruded ink decreases between 50 and 350 °C, which corresponds to the decomposition of the PLGA binder. The atmosphere for debinding is selected as Ar-1 mol.% $O_2$ to prevent cracking caused by binder combustion (Fig. S2). After debinding, another mass loss occurs between 750 and 900 °C, corresponding to the decomposition of $BaCO_3$ releasing $CO_2$. From the in-situ X-ray diffraction data (Fig. 1c), the $BaCO_3$ transforms from the orthorhombic to the trigonal phases at ~700 °C. Thereafter, the tetragonal Y123 phase starts to form, while the decomposition of $BaCO_3$ continues. The diffraction peaks for $BaCO_3$ disappear at ~780 °C, and the peaks of $Ba_{1-y}CuO_{2+\delta} \cdot (CO_2)_x$[26] (Ba-Cu-O-CO$_2$) appear

subsequently, consistent with the TGA results showing that $CO_2$ release is not complete until a temperature of 900 °C is reached. After full $CO_2$ release, the barium copper oxycarbonate is expected to transform to $BaCuO_2$, which forms a transient liquid phase above ~970 °C[27], thus facilitating the densification process. Therefore, a sintering temperature of 1000 °C is selected after comparing with an experiment carried out at 985 °C (Fig. S3). After sintering at 1000 °C for 20 h, the micro-lattice sample exhibits an orthorhombic Y123 phase (Fig. 1d). The micro-lattices, before and after sintering (Fig. 1e), show a uniform linear shrinkage of $34.7 \pm 0.3$ % without cracks or warpage. The densified polycrystal microstructures - with a relative density of $89 \pm 5$ % and a grain size of $1.4 \pm 0.6$ µm - are shown in Fig. 1f.

### Single-crystal growth
For single-crystal growth, the ink composition is altered to achieve a Y-rich, two-phase composition (69 wt.% $YBa_2Cu_3O_{7-x}$ [superconducting Y123] + 30 wt.% $Y_2BaCuO_5$ [non-superconducting Y211] + 1 wt.% $CeO_2$, labeled in the following Y123 + Y211) where (i) the Y211 phase provides additional Y to the Y-depleted melt near the growth front[28] and (ii) $CeO_2$ helps refine the Y211 particles by reacting with Y211 to form $Y_2O_3$, $BaCeO_3$, and CuO[29]. The newly formed $Y_2O_3$ particles can act as nucleation sites for Y211, ultimately leading to the formation of smaller, finely-sized Y211 particles[29], providing more pinning centers for a higher critical current density $J_c$[29,30]. Kim et al. demonstrated that adding 1 wt.% $CeO_2$ to Y123-Y211 can double the critical current density from 1.1 to $2.0 \times 10^4$ A/cm$^2$ at 77 K under a magnetic field of 1 T[30]. After sintering at 1000 °C/ 20 h, the sample shows a densified microstructure with a relative density of $78 \pm 1$ % and a linear shrinkage of $28.3 \pm 0.4$ % (Fig. 2a, b). The sintered micro-lattice is then heated above its peritectic temperature to decompose the Y123 phase to the Y211 + (Ba,Cu-rich) liquid phase[27]. Cooling is performed at a very slow rate (0.5 K/ h) which, upon crossing the peritectic temperature, triggers single-crystal growth initiated from a $NdBa_2Cu_3O_{7-x}$ (NdBCO) single-crystal thin film seed. The formation of a liquid phase during the singe-crystal growth is expected to remove the residual pores by filling the pores and rearranging Y211 particles. A nearly fully densified microstructure is indeed observed (Fig. 2b), consisting of a Y123 matrix in which Y211 and $BaCeO_3$ particles are embedded. Despite liquid formation during the singe-crystal growth, the two printed micro-lattices kept their overall shape and printed details with high fidelity. A small extent of warpage is observed, but it is limited to the 2-3 printed layers (~400 µm) that are close to the seed (Fig. 2e).

Remarkably, the 3D-printed micro-lattice maintains its original shape without slumping or collapsing, despite the large amount of liquid phase needed for single-crystal growth. According to percolation theory, an interconnected 3D solid skeleton starts to form when the volume fraction of jammed hard spheres is above 18.3%[31]. This percolation threshold decreases when the solid particles are spherocylinders[32]: White et al. found that silver nanowires with an aspect ratio of 8 in polystyrene composite have a percolation threshold of 8.3%[32]. The solid Y211 particles exhibit both sphere and cylinder-like shapes on the surface of the lattice (Fig. S4), which is similar to the observation from quenching experiments above the peritectic temperature[30]. If solid Y211 particles are inter-penetrable during sintering, a percolation threshold of 15.6% is obtained for an aspect ratio of 3 for the Y211 particles (Supplementary Information SI1). From the vertical section of the $Y_2O_3$–BaO–CuO phase diagram (Fig. S4), the volume fraction of the solid Y211 phase is ~36% at the temperature of 1090 °C. Therefore, a continuous Y211 skeleton can be expected to form above the peritectic temperature, thus preventing the slumping of the microlattice struts. In addition, various studies of top-seeded infiltration growth[3,11] show that the (Ba,Cu)-rich liquid phase from a bottom liquid source can fully infiltrate upwards into an upper Y211 pellet, driven by capillary forces. Therefore, most of the wetting melt is expected to remain within the porous Y211 skeleton, enabling single-crystal growth without slumping

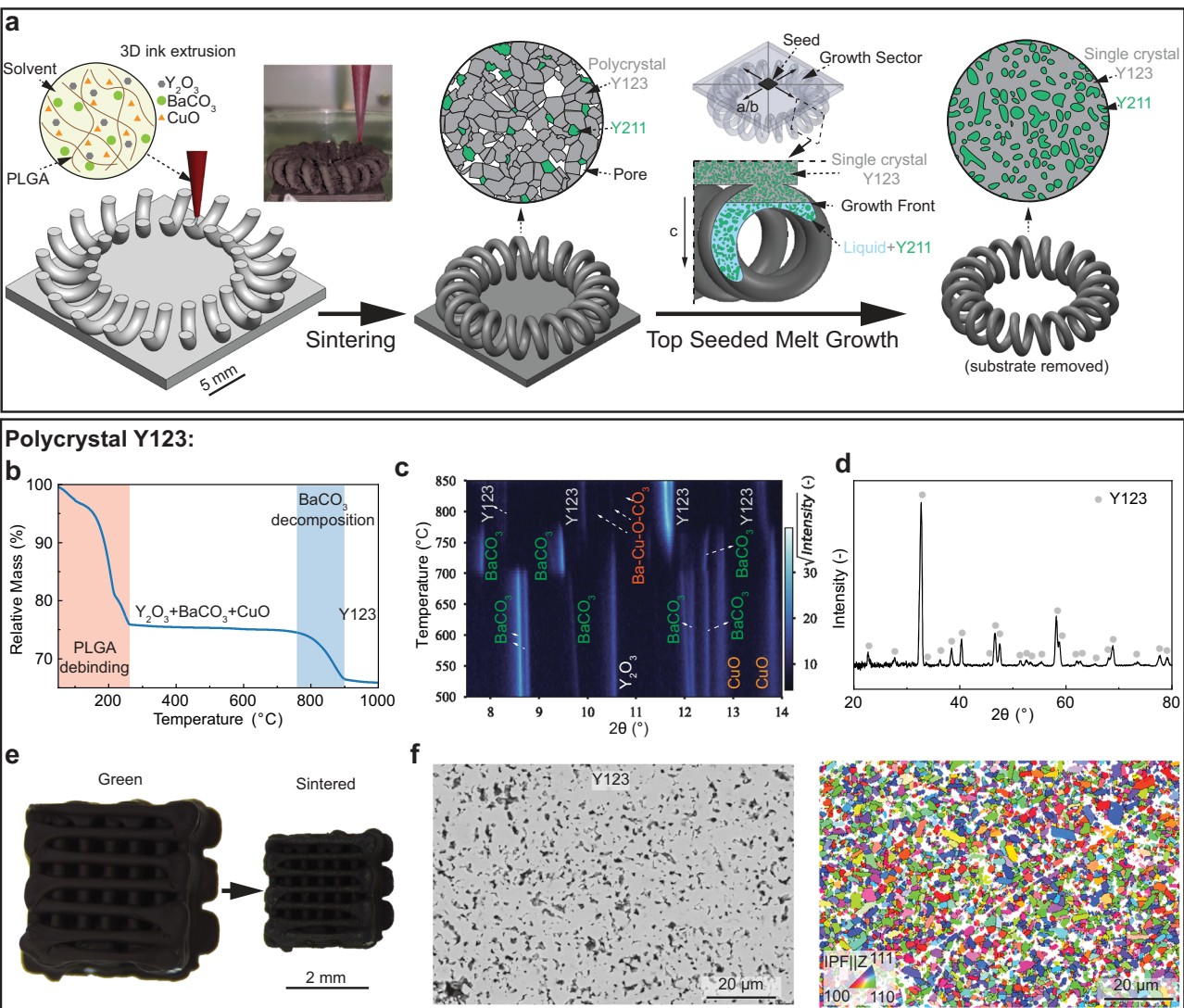

**Fig. 1 | Additive manufacturing route and sintering. a** Schematic illustration of 3D-ink-extrusion printing, sintering, and top-seeded melt growth process of a toroidal YBa$_2$Cu$_3$O$_{7-x}$ (Y123) + Y$_2$BaCuO$_5$ (Y211) coil. (PLGA: poly-lactic-co-glycolic-acid). The inset photograph shows a toroidal coil during layer-by-layer printing from bottom to top, using a 250 μm diameter nozzle. **b** Plot of the relative mass of extruded ink as a function of temperature, as measured by TGA under O$_2$ flow, with two processes (debinding and Y123 synthesis) marked. **c** Stacked X-ray spectra acquired during in-situ diffraction of a blend of Y$_2$O$_3$-BaCO$_3$-CuO upon heating under oxygen from 500 to 850 °C, demonstrating the synthesis of pure Y123. **d** X-ray diffraction spectrum of micro-lattice after sintering at 1000 °C for 20 h. **e** Optical micrographs of the green and the sintered micro-lattice, illustrating uniform shrinkage. **f** SEM micrograph and Inverse Pole Figure map of polished cross-sections of the sintered micro-lattice showing polycrystalline structure.

or collapse of the 3D-printed objects. Reports of 3D printed samples with partial liquid melt are found for other materials, such as liquid sintering in 3D-printed micro-lattices[33,34] and liquid infiltration in 3D-printed porous preforms[35], suggesting that many other materials may be amenable to seeded single-crystal growth after 3D printing.

The XRD patterns shown in Fig. 2c are collected on the top and bottom faces of the micro-lattice. Only the {00 l} family of peaks of the Y123 phase is observed, confirming that the samples are mono-crystalline, with c-axis orientation. Figure 2d displays the IPF map and phase map near the center of the side face of the micro-lattice, showing that small Y211 particles are embedded in the Y123 matrix. The stitched IPF maps on the side face of the micro-lattice are shown in Fig. 2e. As the c-axis is aligned in the vertical direction, the side face shows the {100}/{010} orientation (however, the a and b axes cannot be differentiated from IPF maps due to their very close lattice para-meter). A cubic unit cell (1/3 c) is used for indexing to avoid the mis-indexing problem caused by pseudo-symmetry between the a/b and 1/3 c axis of Y123[36,37]. The stitched IPF maps and pole figure (Fig. 2e) show

that almost the whole lattice is successfully transformed into a single crystal. Three types of defects are also visible. First, some regions with low-angle crystallographic misorientation are visible in the IPF maps near the bottom and edges of the lattice. The complex geometry of the lattice implies that multiple growth fronts advance simultaneously along the struts, with some regions gradually accumulating some crystallographic misorientations. Second, the higher-magnification SEM-BSE micrograph in Fig. 2f shows a few regions of porous Y211 phase located at the middle planes between vertical columns near the top of the lattice, where local growth fronts meet; these are most likely due to local solidification shrinkage of the Y123-forming melt. Lastly, a few horizontal cracks, ending at the porous Y211 planes, are visible in Fig. 2f. They are expected to be caused by different volume expansions in the a/b and c axis and an oxygen concentration gradient during the oxygenation of the Y123 + Y211 micro-lattice. If these cracks adversely impact mechanical and superconductivity properties, they can be mitigated by Ag additions[38], an approach to be explored in future research.

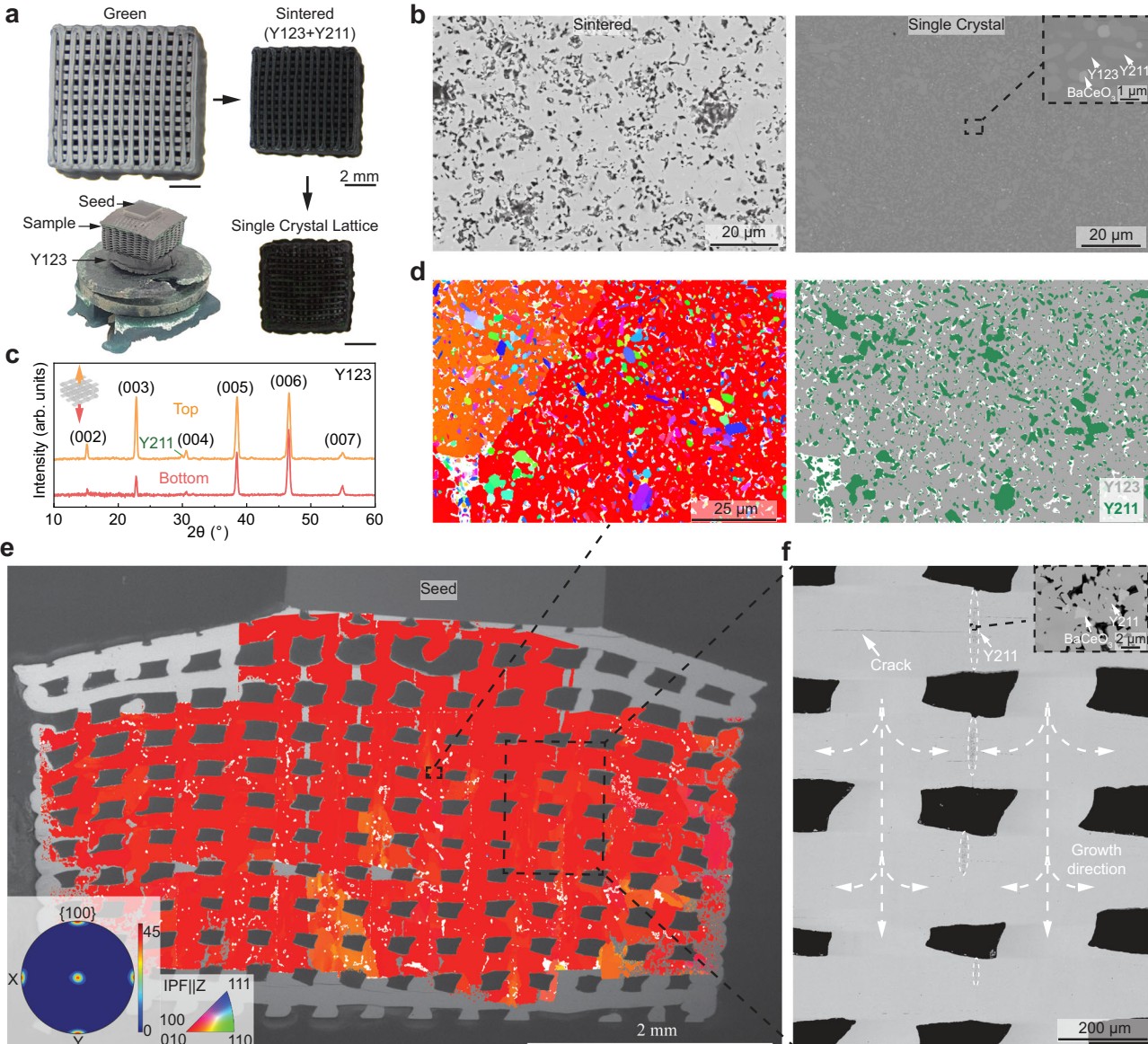

**Fig. 2 | Single-crystal growth. a** Photographs of a micro-lattice after 3D-ink-printing (green state), sintering, and single crystal growth for YBa$_2$Cu$_3$O$_{7-x}$ (Y123) + Y$_2$BaCuO$_5$ (Y211). **b** SEM-BSE micrographs of cross-sections of the 3D-printed lattice after sintering and the top-seeded melt growth, showing efficient removal of pores by the melt. Insert shows Y211 and BaCeO$_3$ particles in the Y123 matrix. **c** XRD spectra for the top and bottom faces of the 3D-printed lattice after single-crystal growth, showing single c-axis orientation on both faces. **d** Higher

magnification IPF (inverse pole figure) map (left) and phase map (right) showing distribution of the Y211 phase (green) in the Y123 phase (gray). **e** Stitched IPF maps of full vertical cross-section on the side face of the 3D-printed lattice, showing {100}/{010} orientation (single crystal), using a cubic version of Y123 for indexing. **f** Enlarged SEM-BSE micrographs showing Y211 concentration at convergence planes (circled), following the growth of the single crystal (marked with dotted lines).

## Superconducting properties

The superconducting properties of sintered ink-ingot samples, including critical transition temperature ($T_c$) from magnetization, resistance measurements, AC susceptibility, and critical current density ($J_c$), are shown in Fig. 3 and Fig. S8 for polycrystalline Y123 and monocrystalline Y123 + Y211. As shown in Fig. 3a, the poly- and monocrystalline samples exhibit nearly the same $T_c$ values ($T_c$ = 89 and 88 K, respectively) from magnetization measurements. The resistance measurement for another monocrystalline sample shows a similar critical transition temperature ($T_c$=89.5 K) in Fig. 3b and has a transition width of 8.5 K. The AC susceptibility measurement for the same monocrystalline sample shows a critical transition temperature of 89 K. These $T_c$ values are about 3.5-5 K below the theoretical value ($T_c$=93 K)[1]. Measurement of impurities show the presence of seven elements - As (0.02 wt.%), B (0.014 wt.%), Ca (0.09 wt.%), Co

(0.01 wt.%), Fe (0.01 wt.%), K (0.01 wt.%), and Ti (0.03 wt.%) – which are known to deteriorate superconducting properties (as summarized in Supplementary Information SI2) and are thus the likely reason for this lowered $T_c$ value. Additively manufactured polycrystalline samples from other studies have shown a range of 86.5 K to 92 K for $T_c$[5,19,20,22–24], based on magnetization or transport measurements. Our single-crystal samples, with a $T_c$ of 88–89.5 K, exhibit a similar critical temperature. The magnetization measurements (Fig. 3a) reveal that the single-crystal sample exhibits a transition temperature about 1 K lower than that of the polycrystalline sample. One likely mechanism explaining this discrepancy is as follows. During the single-crystal growth process, the sample is held at 1090°C for 1 hour, before cooling. This high-temperature step allows for the homogenization of possibly segregated impurities through accelerated diffusion in the liquid phase, which may facilitate the incorporation of some of the seven detected

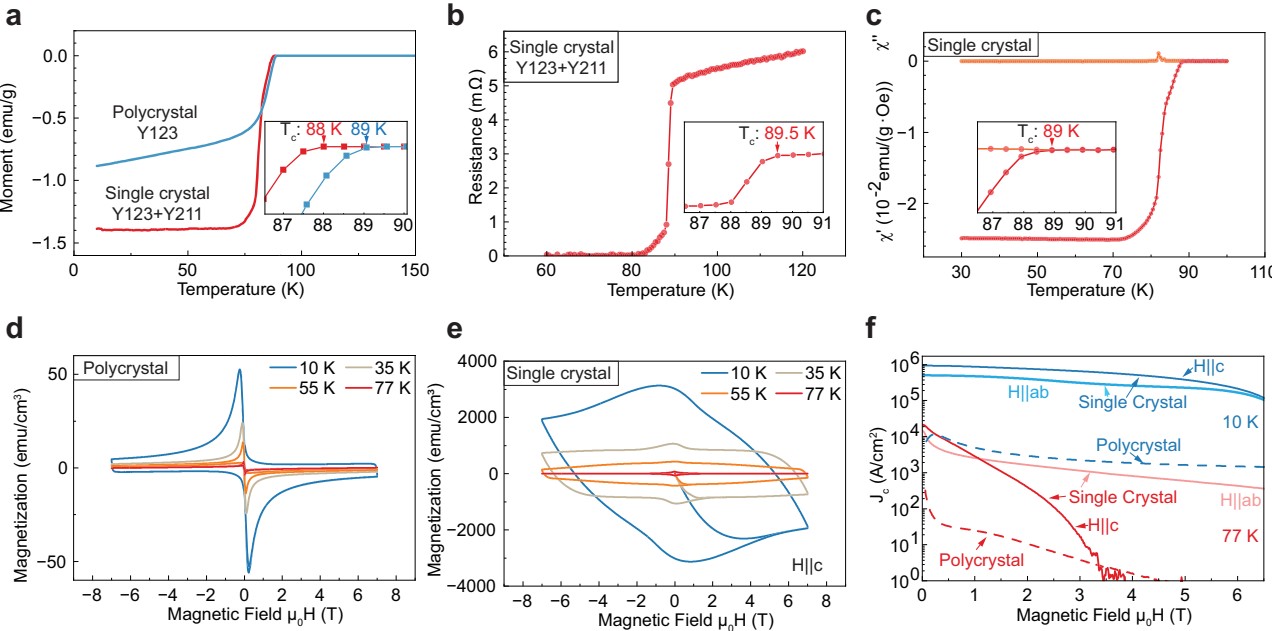

**Fig. 3 | Superconducting properties.** Superconducting properties of YBCO (polycrystalline YBa$_2$Cu$_3$O$_{7-x}$ (Y123) and monocrystalline Y123 + Y$_2$BaCuO$_5$ (Y211)) sintered ink-ingot specimens. **a** Temperature dependence of magnetic moment, as measured at 5 mT and in a zero-field cooling mode, with critical temperatures marked. **b** Temperature dependence of resistance for monocrystalline Y123 + Y211, as measured under zero external magnetic field, with critical temperature marked. **c** Temperature dependence of AC susceptibility of the real and imaginary parts of the magnetic susceptibility ($\chi'$ and $\chi''$) for monocrystalline Y123 + Y211, as measured under AC field amplitude of 0.01 mT at a frequency of 10 Hz. **d**, **e** Plots of magnetization vs. magnetic field at 10, 35, 55, and 77 K (the magnetic field (H) is parallel to the c-axis of the monocrystalline sample, H||c). **f** Plot of critical current density vs. magnetic field at 10 and 77 K (with two orientations for the single crystal: H||c and H||ab).

impurities into the crystal structure. Upon cooling, certain impurities may be expelled near the melt front; however, some residual impurities remain and are incorporated uniformly throughout the sample, which can lead to a slight reduction in the single crystal sample's critical temperature (T$_c$). The presence of small side peaks in the plot of the imaginary part of the complex magnetic susceptibility $\chi''$ below T$_c$ is consistent with the presence of imperfections, which may arise from regions with reduced superconducting transition regions because of impurities, misorientation, or non-uniform oxygenation[39,40]. Figure 3d, e shows the magnetization of polycrystal (pure Y123) and single-crystal samples, respectively. The polycrystal sample shows a much lower magnetization than that of the single-crystal sample. The width of the magnetization hysteresis loop in the vertical direction is related to critical current density, which can be used to calculate critical current density J$_c$ according to the Bean model[41,42]. The calculated values are plotted in Fig. 3f, which shows that the critical current density of single-crystal samples ($2.1 \times 10^4$ A.cm$^{-2}$, H||c) is ~66 times higher than that of polycrystal samples ($3.2 \times 10^2$ A.cm$^{-2}$) at 77 K for a near zero field, consistent with previous 3D-printed polycrystal samples (~$5 \times 10^1$ A.cm$^{-2}$, 77 K)[5] and top seeded melt growth samples (~$4.5 \times 10^4$ A.cm$^{-2}$, 77 K, H||c)[6]. At 10 K, the J$_c$ of single-crystal samples ($9.3 \times 10^5$ A.cm$^{-2}$, H||c) is ~180 times higher than that of polycrystal samples ($5.2 \times 10^3$ A.cm$^{-2}$) for a near zero field. Effective elimination of the grain boundaries contributes to this high critical current density. Anisotropic behavior is also observed when the magnetic field is parallel with the c and a/b axis, as also reported in a directional solidified sample[43]. The Y123-Y211 single crystal produced by the directional solidification method can exhibit a critical current density of $8.8 \times 10^4$ A.cm$^{-2}$ at 77 K with H||c in the self-field[29]. This is of the same order of magnitude as the current density measured in our study. An epitaxial Y123-BaZrO$_3$ thin film (200–270 nm) produced by chemical solution deposition on an SrTiO$_3$ single crystal can achieve a critical

current density of $4.5 \times 10^6$ A.cm$^{-2}$ at 77 K with H||c in the near self-field. The higher critical current density of the thin film is primarily attributed to its better c-axis alignment[44] and a higher density of pinning centers[45]. It should be noted that the tested samples are taken from an ink ingot after sintering and single-crystal growth, as shown in Supplementary Fig. S7. The microlattice architectures or other complex structures may exhibit more misorientation compared to the ink ingot due to their intricate single-crystal growth paths, and may thus show critical current density lower than that shown in Fig. 3f. Accurately measuring the upper critical magnetic field H$_{c2}$ at low temperatures for the above samples would require a higher-field magnet, which exceeds the capabilities of the current facilities.

## 3D-printed objects with complex architectures
The ability to grow monocrystalline YBCO (Y123 + Y211) superconductors from complex 3D-ink-printed polycrystalline parts is demonstrated in Fig. 4a–c. First, a horizontal coil with a closed loop is printed (Supplementary Movie 1), sintered, and transformed into a single crystal (Fig. 4a). The horizontal coil was printed on top of a printed substrate that has the same Y123-Y211 composition. After sintering, the entire sample, including the horizontal coil and substrate, is flipped so that the substrate faces upward. Then, the seed is placed on top of the substrate to allow subsequent single-crystal growth. The faceted growth sectors expand over the whole seeded surface, indicating complete single-crystal growth (Fig. S5). After removing the substrate by grinding, the final 3D-ink-printed monocrystalline part is obtained. The IPF and Phase maps on the side view show that some surface regions with green color are from the Y211 phase and are localized near the sample surface (~50 μm). The IPF map and pole figures (Fig. S6) on the top view confirm that the single crystal grew from top to bottom. After immersing in liquid nitrogen, the coil can be levitated, consistent with T$_c$ > 77 K (Supplementary Movie 2). A

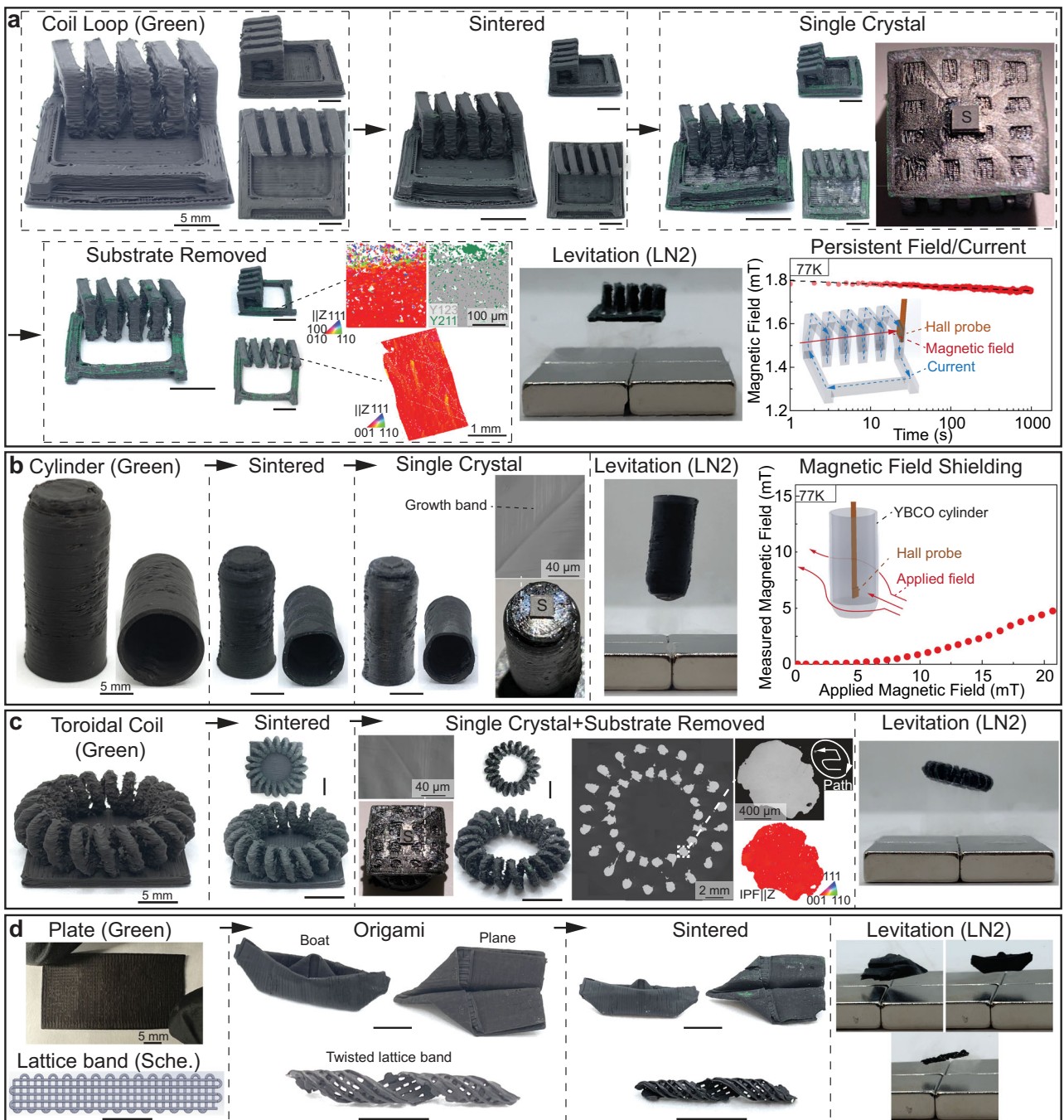

**Fig. 4 | 3D-printed objects with complex architectures.** 3D-printed poly- and monocrystalline objects with complex architectures. **a** Photographs for the 3D-printed YBa$_2$Cu$_3$O$_{7-x}$ (Y123) + Y$_2$BaCuO$_5$ (Y211) horizontal coil loop after printing (green), sintering, single-crystal growth (seed marked with "S"), substrate removal, and levitation at 77 K (LN2: liquid nitrogen). The IPF and phase map on the side view and the IPF map on the top view are added next to the substrate-removed sample. After inducing a persistent field/current, the evolution of the generated magnetic field as a function of time is shown for up to 1000 s. **b** Photographs for the 3D-printed, sintered, monocrystalline, and levitated tube. SEM-BSE micrograph on the seeded surface is shown. The magnetic field as measured inside the tube as a function of the applied outside magnetic field is shown. **c** Photographs for the 3D-printed, sintered, monocrystalline (substrate removed), and levitated toroidal coil. SEM-BSE micrograph on the seeded surface is shown. SEM-BSE micrograph and IPF map on the cross-section show high densification, with individual ink-deposited strands, fused to each other. The printing path is illustrated. **d** Photographs for 3D-printed green plate and schematic figure of green lattice band. The following photographs show a boat, a plane, and a lattice band after Origami folding, sintering (without subsequent single-crystal growth), and levitation.

demonstration of the coil's ability to sustain a persistent current is also demonstrated at 77 K: the current is induced by switching off an external magnetic field (40 mT) along the axis of the horizontal coil (Fig. S9). A magnetic field of 1.8 mT generated from the persistent current (9.5 A or ~560 A·cm$^{-2}$, Supplementary Information SI3) is then detected and decays very slowly with time, following a logarithmic law

(Supplementary Information SI3). It should be noted that current may flow locally across the coil's cross-section; thus, the coil's persistent current may be lower than 9.5 A. The measurement was extended to 10,000 s (Fig. S9c).

Second, Fig. 4b shows a 3D-ink-printed tube, with a wall thickness of 600 µm and a closed-end (Supplementary Movie 3), designed as a

prototype for a shield protecting devices sensitive to external magnetic fields. Growth bands (or growth striations) indicative of singe-crystal growth[46] are observed on the tube surface. At 77 K, the magnetic field (B) within the tube is found to be effectively shielded below 6 mT ($B_{Applied}$ /$B_{Measured}$ > 28). An improved shielding performance could be obtained at lower temperatures, a thicker wall and/or a greater tube length[47,48].

Figure 4c demonstrates a 3D-printed toroidal coil, illustrating the versatility of our 3D-ink-printing method, as this architecture would be extremely difficult to achieve by other methods (Supplementary Movie 4). Multiple closely printed strands form each layer of the looping wire. Single-crystal growth above the peritectic temperature enables full densification, with neither slumping nor void formation between these strands, as observed from SEM micrographs of cross-sections. Many 3D-printed architectures, including coils, may originally not have appropriate seed crystal placement planes. However, Fig. 4a, c demonstrates a route where these architectures still can be made monocrystalline by printing them on top of substrates (Y123-Y211), which are removed after single crystal growth. Therefore, our approach can be applied to most complex shapes.

Finally, as illustrated in Fig. 4d for three miniature objects (airplane, boat, and twisted band), Origami folding of 3D-ink-printed lattice sheets or plates[14] (Supplementary Movie 5) can easily be achieved because the PLGA-DCM ink system is mechanically flexible[25]. After sintering, these polycrystalline demonstration objects retain their shape and levitate at 77 K above permanent magnets (Supplementary Movie 6). The Origami (and related Kirigami) methods can be used to further increase the complexity of 3D-ink-printed samples, with the possibility of "fusing" overlapping, cut edges *via* addition of solvent[14].

This research shows a route combining two seemingly incompatible steps: (i) 3D-ink extrusion printing + sintering and (ii) single-crystal growth. This unlocks design freedom for single-crystal YBCO bulk objects and is applicable to various rare-earth (La, Sm, Nd, Gd, Eu) barium copper oxide compositions. The underlying mechanism is explained by percolation theory where a percolated solid skeleton (Y211) supports the structure's integrity. This work demonstrates the compatibility of semi-solid single-crystal growth with additive manufacturing, suggesting a potential path toward a more universal method. Additionally, this approach could be explored and adapted to the additive manufacturing of other monocrystalline functional materials e.g., piezoelectrics[49], thermoelectrics[50,51], photovoltaics[52], and organic semiconductors[53].

## Methods
### Ink preparation and 3D printing
Inks with a composition of 69 wt.% Y123 + 30 wt.% Y211 + 1 wt.% $CeO_2$ are made by mixing $Y_2O_3$ (1.17 g, purity: 99.995%, powder size: 0.5-1 μm, supplier: SkySpring Nanomaterials, Inc.), $BaCO_3$ (2.37 g, 99.8%, 0.8 μm, US Research Nanomaterials, Inc.), CuO (1.32 g, > 99.95%, 25-55 nm, US Research Nanomaterials, Inc.), $CeO_2$ (0.04 g, 99.97%, 10-30 nm, US Research Nanomaterials, Inc.) powders, with addition of (i) binder: poly-lactic-co-glycolic-acid (0.5 g, PLGA, Evonik Industries), (ii) solvent: dichloromethane (10 ml, DCM, Sigma-Aldrich), (iii) plasticizer: dibutyl phthalate (0.91 g, DBP, Sigma-Aldrich), and (iv) surfactant: ethylene glycol butyl ether (0.45 g, EGBE, Sigma-Aldrich). The composition for pure Y123 inks uses 0.75 g $Y_2O_3$, 2.58 g $BaCO_3$, and 1.56 g $CeO_2$. Roller milling (70 rpm) with zirconia balls (ball to powder ratio: 5:1) is carried out in high-density polyethylene (120 ml) bottle for 48 h to mix powders with DCM and EGBE (Fig. S1). Then, PLGA and EGBE are added for another 12 h ball milling. The excess DCM in the ink is evaporated at ~40 °C to adjust the viscosity with a final powder loading of ~37% (the rheological properties of the PLGA-DBP-DCM ink system have been studied in detail in ref. 14).

3D-ink-printing is carried out with a 3D-Bioplotter (EnvisionTEC, Germany) with conical 250 μm plastic nozzles (Nordson EFD). The

$10 \times 10 \times 5$ mm³ lattices are printed with a layer height of 200 μm, a horizontal spacing of 500 μm, and a layer rotation angle of 90°. The horizontal coil is designed based on Thingiverse.com (thing: 1493458) and has a height of 11 mm, a single contour layer, a layer height of 200 μm, and a horizontal spacing of 350 μm. The toroidal coil is designed with a bending radius of 2.5 mm, a wire diameter of 1.4 mm, and a total turn of 20. The hollow cylinder has a height of 30 mm, a diameter of 14 mm, and two contour layers. The Origami samples, including boat and plane, are folded manually from plates with a thickness of two layers and a horizontal spacing of 0.35 mm. The lattice band sample is bent manually from lattice plates with a thickness of two layers and a horizontal spacing of 650 μm. Lastly, a uniform ink ingot is made by extruding ink (with a decreased level of DCM evaporation) into a cylinder mold followed by drying, which is used for measuring superconducting properties.

### Sintering and growing single crystals
The printed samples are firstly heated under flowing 1 mol.% $O_2$ /Ar for de-binding. Two steps are included in the de-binding: (i) evaporation of residual DCM solvent and EGBE at 150 °C for 30 min, and (ii) decomposition of PLGA binder polymer at 300 °C for 30 min. The heating and cooling rates are 2 °C/ min. Then, the solid-state synthesis and sintering are performed in two steps under flowing $O_2$ (99.999% pure). First, the sample was heated at 880 °C for 10 h with a heating rate of 1 °C/ min. Second, the samples were heated up to 1000 °C for 20 h with a heating rate of 10 °C/ min and a cooling rate of 1 °C/ min.

After sintering, the top seed melt growth is performed to transform 3D-printed polycrystal samples into single-crystal samples. The monocrystalline seed (from Ceraco ceramic coating GmbH) is a $2 \times 2$ mm² MgO substrate, with a 20 nm YBCO thin film and a 500 nm NdBa$_2$Cu$_3$O$_{7-x}$ (NdBCO) thin film. The sintered samples are heated at 1090 °C for 1 h with a heating rate of 100 °C/ h[54], cooled to 1008 °C with a cooling rate of 50 °C/ h, cooled to 991 °C with a cooling rate of 0.5 °C/ h, held at 991 °C for 10 h, and cooled to room temperature with a cooling rate of 1 °C/ min. The atmosphere is under dry air flow. The flipped toroidal coil sample was supported by a sintered Y123 + Y211 tube (diameter: 5.4 mm, height: 3.8 mm) in its center. Samples with the substrate are then fully mounted in Crystalbond 509 (AREMCO) followed by careful grinding on SiC grinding papers. After removing the substrate, the acetone is used to dissolve the Crystalbond. Long-term annealing at 450 °C for 150 h under pure oxygen (200 sccm) is followed for higher oxygen content in Y123.

### Characterization
Thermogravimetric analysis (TGA) is carried out on Mettler Toledo TGA/DSC 3 + . The extruded ink is placed in alumina crucibles. The TGA is then performed up to 1000 °C with a heating rate of 2 °C/ min under oxygen flow (50 ml/min).

In situ X-ray diffraction is performed on STADI MP. The measurement is performed with an Ag source on the $Y_2O_3$-$BaCO_3$-CuO powder blend (target composition: YBa$_2$Cu$_3$O$_{7-x}$) lodged in an externally-heated quartz capillary (0.5 mm ID). Diffraction spectra are detected at each 5 °C step with 18.54° 2θ coverage and 300 s exposure. The heating rate is 5 °C/ min. The ex-situ X-ray diffraction is performed on the cross-sections of sintered samples on a Smartlab 3 kW Gen2.

Optical micrographs of samples are taken by a stereo microscope. Metallographic characterization is done on polished cross-sections of 3D-printed samples. Samples are cold mounted in epoxy resin, ground with SiC grinding papers with ethanol, polished with ethanol-based diamond suspensions (3 and 1 μm) and colloidal silica (0.05 μm), and coated with 4–8 nm Os. Scanning electron microscopy (SEM) and Electron Backscatter Diffraction (EBSD) are performed on a Quanta 650 instrument with an Oxford Symmetry 2 detector.

Superconducting properties including magnetization and resistivity are measured on a PPMS (Physical Properties Measurement

System, Quantum Design) with a Vibrating Sample Magnetometer. The magnetization measurement is performed between 5 K and 150 K under a magnetic field of 5 mT. The resistance measurement is performed between 60 and 120 K at a heating rate of 0.5 K min$^{-1}$ with an applied current of 200 μA under zero external magnetic field. Four probes are attached to the ground surface of the sample by silver paste. The resistance test was conducted with the current flowing along the a/b-axis of sample. The resistivity is not provided here because the half-cylinder shape of the tested sample (with a diameter of 3.1 mm and a thickness of 1.1 mm) does not allow for a consistent cross-sectional area needed for calculation. The AC susceptibility measurement is performed using an MPMS3 (Magnetic Properties Measurement System, Quantum Design) in the temperature range of 30–100 K under zero-field cooling conditions. The measurement is conducted with an AC field amplitude of 0.01 mT applied along the c-axis of the single-crystal sample, zero DC field, and a frequency of 10 Hz. Magnetic hysteresis experiments are performed at 10, 35, 55, and 77 K under a magnetic field between −7 and 7 T. The critical current density $J_c$ (A·cm$^{-2}$) is calculated by Bean model[41,42]:

$$J_c = \frac{20\Delta M}{b\left(1 - \frac{b}{3a}\right)} \tag{1}$$

where the $\Delta M$ (emu cm$^{-3}$) is the width of the hysteresis loop at a specific magnetic field, and $a$ and $b$ (cm) are the dimensions of the cross-section ($a > b$) perpendicular to the applied field.

For the levitation, the sample is cooled in liquid nitrogen and then taken out to place on four Nd-Fe-B magnets ($19 \times 19 \times 6$ mm$^3$, surface field: 0.3 T, K&J Magnetics, Inc.). For the persistent current of the coil loop, an electromagnet with a magnetic field of 40 mT is placed along the axis of the coil (Fig. S9a). The external magnetic field is switched off after the sample is cooled down by a liquid nitrogen bath. A hall probe (EGHKCMS006, PARAGRAF) is used to measure the magnetic field generated by persistent current as a function of time, where each data point is an averaged value from 1000 times of measurements for 1 second. For the magnetic shielding of the cylinder, the cylinder sample with a hall probe inside it is immersed in a liquid nitrogen bath, where an external magnetic field is ramped up step by step (-1 mT /10 s) by an electromagnet next to the sample (Fig. S9b).

## Data availability

The data generated or analyzed during the current study are accessible from the corresponding authors. Source data is provided with this paper. Source data are provided with this paper.

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

## Acknowledgements

This research received funding (D.Z., C.B., and D.C.D.) from the Fermi National Accelerator Laboratory (Fermilab), a U.S. Department of Energy, Office of Science, Office of High Energy Physics HEP User Facility. Fermilab is managed by Fermi Forward Discovery Group, LLC, acting under Contract No. 89243024CSC000002. It made use of Northwestern University's MatSCI and MLTOF facilities that received support from the MRSEC program (NSF DMR-1720139, NSF DMR–2308691, respectively) of the Materials Research Center. This work made use of: (i) the IMSERC Crystallography facility at Northwestern University, which has received support from the Soft and Hybrid Nanotechnology Experimental (SHyNE) Resource (NSF ECCS–2025633), and Northwestern University; (ii) the Jerome B. Cohen X-Ray Diffraction Facility supported by the MRSEC program of the National Science Foundation (DMR–2308691) at the Materials Research Center of Northwestern University and the Soft and Hybrid Nanotechnology Experimental (SHyNE) Resource (NSF ECCS-1542205.); (iii) the EPIC facilities at NUANCE center that received support from Soft and Hybrid Nanotechnology Experimental (SHyNE) Resource (NSF ECCS–2025633), the International Institute for Nanotechnology (IIN), the MRSEC program (NSF DMR–2308691) at the Materials Research Center, the Keck Foundation, and the State of Illinois, through the IIN. This work made use of a Quantum Design MPMS–3 at the Cornell Center for Materials Research shared instrumentation facility. The authors thank Mr. Steve Kriske from Cornell University for conducting the AC susceptibility tests. We thank Prof. Samuel Stupp (NU) for letting us use his Bioplotter for ink printing. We thank, for their experimental help and/or useful discussions, Prof. Sumit Kewalramani (XRD) and Prof. Chris D. Malliakas (in situ XRD). We acknowledge Ceraco Ceramic Coating GmbH (Ismaning, Germany) for providing the NdBCO seeds.

## Author contributions

D.Z. and D.C.D. proposed the concept and designed the experiments. D.Z. performed 3D printing, thermal treatments, and characterization experiments. D.Z., C.B., and D.C.D. analyzed data and discussed the findings. D.Z. wrote the first manuscript. D.Z., C.B., and D.C.D. contributed to the manuscript writing.

## Competing interests

A provisional patent application covering this work has been filed by Northwestern University (U.S. Provisional Patent Application No. 63/550,705), with D.Z., C.B., and D.C.D. as inventors and applicants. A U.S. non-provisional (utility) application based on this provisional application is being prepared. D.C.D. discloses a financial interest in Metalprinting, Inc. (South Korea) which is active in ink-based materials printing.
