## [Transparent Peer Review file · Nature Communications]

Additively-manufactured monocrystalline YBCO superconductor

Corresponding Author: Dr Dingchang Zhang

Version 0:

Reviewer comments:

Reviewer #1

(Remarks to the Author)

In this work, the authors studied a additively-manufactured method to produce monocrystalline YBCO superconductor with complex shapes by 3D-ink-printing. they successfully manufactured YBCO with various architectures. While the work showcases a new way of producing toroidal coil, micro-lattice, horizontal coil loop, levitated tube, Origami folding and Twisted structure based on 3D-ink-extrusion printing technology. Although this idea is interesting, its scientific research value and application value are insufficient. It can not be considered.

1. How to control the rheological properties of the slurry? It is difficult to achieve high-precision complex shapes without precise control.
2. How to form a toroidal coil using 3D-ink-extrusion printing method without supporting auxiliary materials, and how to control the accuracy and size?
3. How to ensure the accuracy of superconducting characterization for many structures and shapes?
4. The shrinkage of printed samples is significant, which has a great impact on the complex structure.
5. The grain orientation is studied through EBSD experiments for YBCO on a large scale in Figure 2. The correctness of this decision may be doubted.
6. The prepared devices were still at a relatively low level of current carrying capacity, which is no perspective for applications.
7. The preparation method remains at the level of extrusion printing without any breakthroughs in updating the hierarchy.

Reviewer #2

(Remarks to the Author)

The authors have addressed the points I raised in my initial review to a satisfactory extent. The paper is now suitable for publication in Nature Communications.

Reviewer #4

(Remarks to the Author)

I was invited to comment on the authors' revision in response to the concerns raised by Reviewer #3. While the revised manuscript shows improvement, I still have concerns regarding the responses to some of Reviewer #3's questions. I would like further clarification on these issues before making a final decision on the manuscript's suitability for publication. These concerns are significant because the authors are making an important claim regarding the growth of single-crystalline YBCO superconductors through additive manufacturing. A more thorough and meticulous comparison is required to convincingly demonstrate the single crystallinity of the samples.

I am not satisfied with the authors' response to Comment 3 from Reviewer #3 and have the following concerns that need to be addressed:

1. In the newly added Figure 3(b), why have the authors not presented the R-T measurement for the polycrystalline samples? How does the performance compare to that of the polycrystalline samples?
 2. Following up on the first point, it appears that in the new plot 3(b), the Residual Resistivity Ratio (RRR) is very low. This does not adequately address the previous concern regarding non-uniformity and structural defects in the superconducting sample.
 3. The authors state that "the resistivity is not provided here because the half-cylinder shape of the tested sample does not allow for a consistent cross-sectional area needed for the calculation." However, it remains unclear how much area was actually probed with the four-point measurement. Is the probed area sufficiently large to ensure that they are not only examining local structures, but rather capturing the overall sample characteristics?
 4. Presenting the susceptibility measurement data would provide cross-confirmation and ensure consistency with the other measurements.
 5. In Figure 3(a), the polycrystalline sample shows a higher T_c than the single crystal, whereas the opposite is typically expected. What was the least count of the thermometers used, and how do the authors explain this discrepancy?
 6. In addition, the comparison of H_{c2} between the samples has not been provided. This comparison would be valuable for assessing differences between the single-crystal and polycrystalline samples.
- Those were the major concerns. Aside from these, the authors have addressed the concerns raised by Reviewer #3, with the exception of one minor issue.

7The claim regarding the microstructure's contribution to higher performance remains unconvincing. I suggest that the authors adjust the tone of the manuscript to more accurately reflect the implications of the microstructure on performance. In summary, I do not recommend publishing the manuscript without addressing the concerns outlined above. If the authors adequately address these issues, I believe the manuscript would be suitable for publication in Nature Communications.

Version 1:

Reviewer comments:

Reviewer #1

(Remarks to the Author)

The authors have addressed the points I raised in my review to a satisfactory extent. The paper is now suitable for publication in Nature Communications.

Reviewer #4

(Remarks to the Author)

Regarding the authors' responses to the previous comments, I am satisfied with the clarifications provided for comments #1, #2, and #3. However, I still have reservations about the necessity of susceptibility measurements, as these are fundamental characterizations for superconducting materials. Incorporating such measurements would provide a more comprehensive validation of the claims.

Additionally, I did not observe an explanation for the discrepancy in the critical temperature (T_c) noted in comment #5 within the updated manuscript. I suggest that the authors include a discussion of potential reasons for this discrepancy to strengthen the manuscript further.

Lastly, while I understand that performing H_{c2} measurements may be beyond the current experimental capabilities, acknowledging this limitation explicitly in the manuscript would be appropriate and acceptable.

Once these three concerns are addressed, I believe the manuscript will be suitable for publication in Nature Communications.

Version 2:

Reviewer comments:

Reviewer #4

(Remarks to the Author)

Authors have addressed all the concerns that I had raised before and I recommend publishing in this updated version.

Dear Editor,

We appreciate the three reviewers' time to review our manuscript and their valuable comments, as well as your kind decision to give us this opportunity to respond to them. We revised the paper in accordance with all their comments, and we believe that our revisions fully address all the concerns raised by the reviewers and significantly enhance the quality of the manuscript.

Below is a detailed response to their comments.

Sincerely,

Dingchang Zhang

For **Reviewer #1**:

“In this work, the authors studied a additively-manufactured method to produce monocrystalline YBCO superconductor with complex shapes by 3D-ink-printing. they successfully manufactured YBCO with various architectures. While the work showcases a new way of producing toroidal coil, micro-lattice, horizontal coil loop, levitated tube, Origami folding and Twisted structure based on 3D-ink-extrusion printing technology. Although this idea is interesting, its scientific research value and application value are insufficient. It can not be considered.”

Thank you for your comment and appreciation of the novelty in processing.

Scientifically, we believe our ms brings value as follows. This study successfully combines single-crystal growth with additive manufacturing for the first time. This is quite remarkable, especially since the geometric features of printed parts are preserved, even under prolonged liquid-phase exposure above the peritectic temperature during crystal growth. The scientific underlying mechanism is explained by **percolation theory** where a percolated solid skeleton supports the structure's integrity. This method, because it is science-based and not just a “recipe” is **widely applicable**: to various rare-earth barium copper oxide **superconductors** (La, Sm, Nd, Gd, Eu) and to other semi-**solid single crystal growth process**. It can also be adapted for other monocrystalline functional materials, including **piezoelectric, thermoelectrics, photovoltaics, and organic semiconductors**. A patent about this route has been pursued by Northwestern University.

In order to clarify this point, these sentences are added in page 17 line 291,

“The underlying mechanism is explained by percolation theory where a percolated solid skeleton (Y211) supports the structure's integrity. This work demonstrates the compatibility of semi-solid single-crystal growth with additive manufacturing, suggesting a potential path toward a more universal method.”

In terms of **applications** value, bulk monocrystalline YBCO superconductor have important applications. For example, YBCO superconducting **undulators** for synchrotron or a free-electron laser has been developed by Paul Scherrer Institute very recently. (Calvi, Marco, et al. "Experimental results of a YBCO bulk superconducting

undulator magnetic optimization." *Physical Review Accelerators and Beams* 27.10 (2024): 100702.). Superconducting **levitation** conveyor systems suitable for clean room has been developed and commercialized by Festo's SupraMotion.

Other applications such as superconducting **magnetic bearings for motors and flywheel**, components for maglev trains, **energy storage applications**, superconducting magnetic **shielding** based applications are summarized in Ref [3]: (Namburi DK, Shi Y, Cardwell DA. The processing and properties of bulk (RE)BCO high temperature superconductors: current status and future perspectives. *Supercond Sci Technol* 34, 053002 (2021).)

Additive manufacturing of monocrystalline YBCO superconductors not only provide more design freedom for these existing applications but also allow for **brand new applications**. **Shapes** of bulk YBCO are important to obtain **optimized magnetic field distribution, efficient cooling, and low mass** for levitation, which can be achieved easily by additive manufacturing.

In order to clarify this point, below sentences are added in page 2 line 40,

“Additive manufacturing of monocrystalline YBCO superconductors not only offers greater design flexibility for existing applications but also paves the way for new innovations. The architectural freedom of additive manufacturing enables optimized magnetic field distribution, efficient cooling to sustain the superconducting state, and a lightweight structure for levitation applications. Additive manufacturing of ceramics, like YBCO, currently produces only a polycrystalline microstructure when followed by conventional sintering.”

1. “*How to control the rheological properties of the slurry? It is difficult to achieve high-precision complex shapes without precise control.*”

Thank you for raising this concern. The rheological properties can be easily controlled by adjusting the solvent (DCM) **evaporation time** in a water bath maintained at a constant temperature of 40 °C. The rheological properties of the PLGA-DBP-DCM ink system have been studied in detail in Ref. [12].

2. “*How to form a toroidal coil using 3D-ink-extrusion printing method without supporting auxiliary materials, and how to control the accuracy and size?*”

Thank you for this point. **No supporting auxiliary materials** are used when printing a toroidal coil, as shown in Figure 1(a).

To clarify this point, below sentences are added in page 6 line 94:

“The rapid solvent evaporation after ink extrusion causes the binder to precipitate immediately after extrusion thereby increasing the strength of the deposited material, unlike non-evaporating ink systems. As a result, this ink system does not show slumping and sagging of the printed filaments²⁵ and thus enables them to partially overhang and form the arcs in the toroidal coil **without auxiliary supports.**”

High accuracy in printed structures can be achieved by using a **precise printing head and fine nozzle sizes**. Predictable, **reproducible shrinkage** can be accounted for to ensure accurate dimensions after sintering. This approach is **routinely applied in the production of sintered ceramics**.

3. *“How to ensure the accuracy of superconducting characterization for many structures and shapes?”*

We appreciate your raising this point. For various structures and shapes, the accuracy of superconducting characterizations for these intensive properties is controlled by the following aspects:

- (1) Superconducting characterization itself is accurate by using PPMS system (Physical Measurement System, Quantum Design).
- (2) We ensured **homogeneous mixing of precursor powders** by extended ball milling and verified this with EDS, as shown in Supplementary Figure 1, confirming uniform composition for different printed structures.
- (3) Our T_c characterizations, conducted on three different samples (Figure 3(a) and (b)), showed a narrow range of 88 K to 89.5 K, demonstrating **good consistency across samples**. These measurements for intensive properties are **not significantly influenced by the shape** of the samples.
- (4) Various printed structures and shapes, shown in Figure 4, exhibited superconducting behavior (**Meissner effect**) when levitated in liquid nitrogen.
- (5) Multiple EBSD characterizations were performed on different structures and shapes, including lattice, horizontal coil loop, and toroidal coil loop. These characterizations confirmed **good single-crystal growth across these structures**, suggesting close superconducting properties (T_c and J_c) with measurements by following the relationship between composition, microstructure, and property.

4. *“The shrinkage of printed samples is significant, which has a great impact on the complex structure.”*

Thank you for pointing this out. The shrinkage after sintering is **reproducible and consistent** due to the well-mixed submicron powders, allowing accurate dimensions to be achieved by accounting for this shrinkage—a **standard practice in sintered ceramic production**. Although slight warping may occur in certain shapes after sintering and single-crystal growth, this can be resolved through optimized supports, minimized friction with the substrate, and uniform temperature distribution. Thus, shrinkage does not hinder the effectiveness of our current method.

5. *“The grain orientation is studied through EBSD experiments for YBCO on a large scale in Figure 2. The correctness of this decision may be doubted.”*

We appreciate you bringing up this concern. EBSD is a **well-established method routinely used to determine the orientation of single crystals**, including thermoelectrics (e.g., Zhao, Li-Dong, et al. "Ultralow thermal conductivity and high thermoelectric figure of merit in SnSe crystals." *Nature* 508.7496 (2014): 373-377.) and high-entropy perovskite single crystals (e.g., Folgueras, Maria C., et al. "High-entropy halide perovskite single crystals stabilized by mild chemistry." *Nature* 621.7978 (2023): 282-288).

Additionally, we provided **XRD results** from the top and bottom surfaces of our sample in Figure 2(c), where only (00 l) peaks appear for the Y123 phase in the XRD profiles, aligning with the EBSD results. The **combination of XRD and EBSD** confirms successful single-crystal growth.

6. *“The prepared devices were still at a relatively low level of current carrying capacity, which is no perspective for applications.”*

Thank you for this point. Our single crystal sample demonstrates a **comparable** current density ($2.1 \times 10^4 \text{ A}\cdot\text{cm}^{-2}$ at 77K with H||c) **with conventional bulk single crystal samples** ($4.5 \times 10^4 \text{ A}\cdot\text{cm}^{-2}$ at 77K with H||c). Thus, our sample shows good potential for applications that demand **high current densities, such as superconducting trapped field magnets, superconducting undulators, and superconducting levitation conveyor systems**. We believe that our results position this sample as a viable candidate for further development in these advanced applications.

7. *“The preparation method remains at the level of extrusion printing without any breakthroughs in updating the hierarchy.”*

Thank you for pointing out this aspect. In this research, extrusion printing is used as a representative additive manufacturing method, and our experimental results **can readily be extended to other additive manufacturing techniques** to enhance structural complexity and hierarchical organization.

Additionally, we have identified compatible methods for **creating hierarchical structures** for lightweight. Large channels can be incorporated through 3D design, macropores can be introduced using space holders, and micropores can be achieved through partial sintering (Kenel, C., et al. "Hierarchically-porous metallic scaffolds via 3D extrusion and reduction of oxide particle inks with salt space-holders." Additive Manufacturing 37 (2021): 101637).

Reviewer #2 (Remarks to the Author):

The authors have addressed the points I raised in my initial review to a satisfactory extent. The paper is now suitable for publication in Nature Communications.

Reviewer #4 (Remarks to the Author):

I was invited to comment on the authors' revision in response to the concerns raised by Reviewer #3. While the revised manuscript shows improvement, I still have concerns regarding the responses to some of Reviewer #3's questions. I would like further clarification on these issues before making a final decision on the manuscript's suitability for publication.

These concerns are significant because the authors are making an important claim regarding the growth of single-crystalline YBCO superconductors through additive manufacturing. A more thorough and meticulous comparison is required to convincingly demonstrate the single crystallinity of the samples. I am not satisfied with the authors' response to Comment 3 from Reviewer #3 and have the following concerns that need to be addressed:

1. *“In the newly added Figure 3(b), why have the authors not presented the R-T measurement for the polycrystalline samples? How does the performance compare to that of the polycrystalline samples?”*

Thank you for pointing this out. Our paper focuses on single-crystal samples, with resistance measurements providing additional evidence of superconductivity. R-T measurements and other characterizations for polycrystalline samples fabricated by additive manufacturing have been documented in multiple references (Refs. 5, 19, 20, 22, 23, 24). For instance, in Ref. 24 (Coveney A, Morrison K, Engström D. DC Resistance Measurements in Multi-Layer Additively Manufactured Yttrium Barium Copper Oxide Components. Adv Eng Mater, 2300773 (2023)), an R-T measurement for polycrystalline samples shows a T_c of 86.5 K, which is lower than the T_c of 89.5 K observed in our single-crystal samples.

To clarify this point, below sentences are added in page 14, line 220:

“Additively manufactured polycrystalline samples from other studies have shown a range of 86.5 K to 92 K for T_c ^{5, 19, 20, 22, 23, 24}, based on magnetization or transport measurements. Our single-crystal samples, with a T_c of 89.5 K, exhibit a similar critical temperature.”

2. *“Following up on the first point, it appears that in the new plot 3(b), the Residual Resistivity Ratio (RRR) is very low. This does not adequately address the previous concern regarding non-uniformity and structural defects in the superconducting sample.”*

The previous concerns: *“In particular, the samples shown in the manuscript has T_c less than the optimal doping level of 93 K, which can be caused by the defect formation or non-uniformity of superconductivity. Transport can shed light on the cause.”*

This is an insightful point. Residual Resistivity Ratio (RRR) is commonly used as an indirect measure of material purity. Rather than relying on this approach, we **directly assessed impurity levels** through Inductively Coupled Plasma (ICP) testing, identifying the presence of seven impurity elements—As, B, Ca, Co, Fe, K, and Ti—all of which are known to degrade superconducting properties (details in Supplementary Information SI2). We believe these impurities are the primary reason for the observed T_c being lower than the ideal 93 K value.

Regarding concerns about non-uniformity and structural defects: Two single-crystal samples from different locations (Supplementary Figure S7) exhibited critical transition temperatures T_c of 88 K and 89.5 K, indicating slight non-uniformity. This **minor variation is typical in top-seeded melt-grown YBCO crystals** along the growth direction, as noted in Ref: (Namburi, Devendra K., et al. "A novel, two-step top seeded infiltration and growth process for the fabrication of single grain, bulk (RE) BCO superconductors." Superconductor Science and Technology 29.9 (2016): 095010). Although structural defects, such as cracks (shown in Figure 2(f)), are present, they are not responsible for the reduced T_c of our samples."

3. “The authors state that “the resistivity is not provided here because the half-cylinder shape of the tested sample does not allow for a consistent cross-sectional area needed for the calculation.” However, it remains unclear how much area was actually probed with the four-point measurement. Is the probed area sufficiently large to ensure that they are not only examining local structures, but rather capturing the overall sample characteristics?”

Thank you for pointing out this aspect. The probed area has an approximate cross-sectional size of $\sim 3 \times 1 \text{ mm}^2$, with a 2 mm distance between the voltage probes. The sample itself—a half-cylinder with a radius of 3.1 mm and thickness of 1.1 mm, shown in Supplementary Figure S7—is already near the size limit of our PPMS stage. We believe this sample size is sufficient to capture the overall characteristics of the material. Larger-scale superconducting characterization is beyond the capacity of our current equipment.

Supplementary Figure S7. Schematic figure showing an ink ingot after sintering and single-crystal growth. The locations and dimensions of the two samples used for magnetization ($2.2 \times 1.1 \times 1.8 \text{ mm}^3$) and resistance measurements (half cylinder, a radius of 3.1 mm and a thickness of 1.1 mm) are labeled.

4. “Presenting the susceptibility measurement data would provide cross-confirmation and ensure consistency with the other measurements.”

Thank you for your suggestion. Both magnetization and resistance measurements on single-crystal samples confirm the superconductivity of our material, with transition temperatures of 88 K and 89.5 K observed across different samples and methods. These results provide cross-confirmation and demonstrate consistency, despite minor non-uniformity from the top-seeded melt growth process and trace impurities. Given this corroboration, we believe that additional susceptibility measurements will not bring new information.

5. “In Figure 3(a), the polycrystalline sample shows a higher T_c than the single crystal, whereas the opposite is typically expected. What was the least count of the thermometers used, and how do the authors explain this discrepancy?”

Thank you for addressing this point. We used a Physical Property Measurement System (PPMS, DynaCool) and it is accurate at the level of **0.1 K**.

During the single-crystal growth process, the sample is held at 1090°C for 1 hour before cooling. This high-temperature step allows for the homogenization of possibly segregated impurities through accelerated diffusion in the liquid phase, which may facilitate the incorporation of some of the seven detected impurities into the crystal structure. Upon cooling, **certain impurities may be expelled near the melt front**; however, some residual impurities remain and are **incorporated uniformly throughout the sample**, which can lead to a slight reduction in the single crystal sample’s critical temperature (T_c).

6. “In addition, the comparison of H_{c2} between the samples has not been provided. This comparison would be valuable for assessing differences between the single-crystal and polycrystalline samples.”

It is an interesting point. Unfortunately, accurately measuring H_{c2} requiring an extremely strong magnet and is only available in few laboratories, which is outside the capability of our current facility. From the literature shown below, the H_{c2} (or B_{c2}) of their polycrystal sample is comparable to the H_{c2} ($H||c$) but is much lower than H_{c2} ($H||ab$) of single crystal sample.

[REDACTED]

Figure: B_{c2} of (a) a single crystal YBCO sample as a function of temperature for magnetic fields $H||c$ and $H||c$ and (b) a polycrystal YBCO sample

(a: Krabbes, Gernot, et al. High temperature superconductor bulk materials: fundamentals, processing, properties control, application aspects. John Wiley & Sons, 2006.

b: Jha, Rajveer, Poonam Rani, and V. P. S. Awana. "Revisiting heat capacity of bulk polycrystalline $YBa_2Cu_3O_{7-\delta}$." *Journal of Superconductivity and Novel Magnetism* 27 (2014): 287-291.)

Those were the major concerns. Aside from these, the authors have addressed the concerns raised by Reviewer #3, with the exception of one minor issue.

7 *“The claim regarding the microstructure's contribution to higher performance remains unconvincing. I suggest that the authors adjust the tone of the manuscript to more accurately reflect the implications of the microstructure on performance.*

We agree with your concern. Below sentences are added to further explain the effect of single crystal microstructure in Page 3, Line 48:

Therefore, YBCO with a polycrystalline microstructure typically has a low critical current density ($\sim 5 \times 10^1 \text{ A}\cdot\text{cm}^{-2}$, 77K, zero field)⁵ due to the presence of numerous grain boundaries. In contrast, forming a single-crystal microstructure effectively eliminates these grain boundaries, leading to a significant increase in critical current density ($\sim 4.5 \times 10^4 \text{ A}\cdot\text{cm}^{-2}$, 77K, $H||c$, zero field)⁶.

In summary, I do not recommend publishing the manuscript without addressing the concerns outlined above. If the authors adequately address these issues, I believe the manuscript would be suitable for publication in Nature Communications.”

Dear Editor,

We appreciate the two reviewers' time to review our manuscript again and their valuable comments. We revised the paper in accordance with all their comments, and we believe that our revisions fully address all the concerns raised by the reviewers and significantly enhance the quality of the manuscript.

Below is a detailed response to their comments.

Yours sincerely,

Dingchang Zhang

David C. Dunand

TMS Fellow, Fellow ASM

Professor of Materials Science and Engineering

Reviewer #1 (Remarks to the Author):

The authors have addressed the points I raised in my review to a satisfactory extent. The paper is now suitable for publication in Nature Communications.

Thank you for your time and commitment, and for reviewing our changes.

Reviewer #4 (Remarks to the Author):

Regarding the authors' responses to the previous comments, I am satisfied with the clarifications provided for comments #1, #2, and #3.

Thank you for reviewing our changes, we are glad they addressed your earlier concerns.

However, I still have reservations about the necessity of susceptibility measurements, as these are fundamental characterizations for superconducting materials. Incorporating such measurements would provide a more comprehensive validation of the claims.

Thank you for your suggestion. We have carried out susceptibility measurement on our single crystal sample as shown in Figure 3 (c), in Page 13 line 200. As we did not have in-house equipment, we contracted with Dr. Steve Kriske at Cornell University to perform the measurements at Cornell's Materials Research Center. We are now acknowledging Dr. Kriske.

Figure 3. Superconducting properties of YBCO (polycrystalline Y123 and monocrystalline Y123+Y211) sintered ink-ingot specimens. (a) Temperature dependence of magnetic moment, as measured at 5 mT and in a zero-field cooling mode, with critical temperatures marked. (b) Temperature dependence of resistance for monocrystalline Y123+Y211, as measured under zero external magnetic field, with critical temperature marked. (c) Temperature dependence of AC susceptibility of the real and imaginary parts of the magnetic susceptibility (χ' and χ'') for monocrystalline Y123+Y211, as measured under AC field amplitude of 0.01 mT at a frequency of 10 Hz. (d,e) Plots of magnetization vs. magnetic field at 10, 35, 55, and 77 K (the magnetic field (H) is parallel to the c-axis of the monocrystalline sample, H||c). (f) Plot of critical current density vs. magnetic field at 10 and 77 K (with two orientations for the single crystal: H||c and H||ab).

Page 14, line 217:

The AC susceptibility measurement for the same monocrystalline sample shows a critical transition temperature of 89 K.

Page 14, line 235:

The presence of small side peaks in the plot of the imaginary part of the complex magnetic susceptibility χ'' below T_c is consistent with the presence of imperfections, which may arise from regions with reduced superconducting transition regions because of impurities, misorientation, or non-uniform oxygenation^{40, 41}.

Page 24, line 403:

The AC susceptibility measurement is performed using an MPMS3 (Magnetic Properties Measurement System, Quantum Design) in the temperature range of 30–100 K under zero-field cooling conditions. The measurement is conducted with an AC field amplitude of 0.01 mT applied along the c-axis of the single-crystal sample, zero DC field, and a frequency of 10 Hz.

Page 27, line 454, Acknowledgment:

This work made use of a Quantum Design MPMS-3 at the Cornell Center for Materials Research shared instrumentation facility. The authors thank Mr. Steve Kriske from Cornell University for conducting the AC susceptibility tests.

Additionally, I did not observe an explanation for the discrepancy in the critical temperature (T_c) noted in comment #5 within the updated manuscript. I suggest that the authors include a discussion of potential reasons for this discrepancy to strengthen the manuscript further.

Thank you for your suggestion. The explanation is now added in the updated manuscript on page 14, line 226,

“The magnetization measurements (Figure 3(a)) reveal that the single-crystal sample exhibits a transition temperature about 1 K lower than that of the polycrystalline sample. One likely mechanism explaining this discrepancy is as follows. During the single-crystal growth process, the sample is held at 1090°C for 1 hour, before cooling. This high-temperature step allows for the homogenization of possibly segregated impurities through accelerated diffusion in the liquid phase, which may facilitate the incorporation of some of the seven detected impurities into the crystal structure. Upon cooling, certain impurities may be expelled near the melt front; however, some residual impurities remain and are incorporated uniformly throughout the sample, which can lead to a slight reduction in the single crystal sample’s critical temperature (T_c).”

Lastly, while I understand that performing H_{c2} measurements may be beyond the current experimental capabilities, acknowledging this limitation explicitly in the manuscript would be appropriate and acceptable.

Thank you for this point. The below text is added in the updated manuscript on page 15, line 261:

Accurately measuring the upper critical magnetic field H_{c2} at low temperatures for the above samples would require a higher-field magnet, which exceeds the capabilities of the current facilities.

Once these three concerns are addressed, I believe the manuscript will be suitable for publication in Nature Communications.

Thank you again for your thorough review which has clearly improved the quality of our manuscript.